# A Crowd Movement Analysis Method Based on Radar Particle Flow

**DOI:** 10.3390/s24061899

**Published:** 2024-03-15

**Authors:** Li Zhang, Lin Cao, Zongmin Zhao, Dongfeng Wang, Chong Fu

**Affiliations:** 1School of Information and Communication Engineering, Beijing Information Science and Technology University, Beijing 100101, China; zhanglilyj@gmail.com (L.Z.);; 2Key Laboratory of the Ministry of Education for Optoelectronic Measurement Technology and Instrument, Beijing Information Science and Technology University, Beijing 100101, China; 3Beijing TransMicrowave Technology Company, Beijing 100080, China; wdf@tsmtc.com; 4School of Computer Science and Engineering, Northeastern University, Shenyang 110819, China; fuchong@mail.neu.edu.cn

**Keywords:** crowd movement analysis, radar particle flow, millimeter-wave radar, particle flow diffusion potential

## Abstract

Crowd movement analysis (CMA) is a key technology in the field of public safety. This technology provides reference for identifying potential hazards in public places by analyzing crowd aggregation and dispersion behavior. Traditional video processing techniques are susceptible to factors such as environmental lighting and depth of field when analyzing crowd movements, so cannot accurately locate the source of events. Radar, on the other hand, offers all-weather distance and angle measurements, effectively compensating for the shortcomings of video surveillance. This paper proposes a crowd motion analysis method based on radar particle flow (RPF). Firstly, radar particle flow is extracted from adjacent frames of millimeter-wave radar point sets by utilizing the optical flow method. Then, a new concept of micro-source is defined to describe whether any two RPF vectors originated from or reach the same location. Finally, in each local area, the internal micro-sources are counted to form a local diffusion potential, which characterizes the movement state of the crowd. The proposed algorithm is validated in real scenarios. By analyzing and processing radar data on aggregation, dispersion, and normal movements, the algorithm is able to effectively identify these movements with an accuracy rate of no less than 88%.

## 1. Introduction

Crowd movement analysis (CMA) is a crucial technology in the intelligent security field [1]. CMA can assist in capturing abnormal events by extracting and analyzing the motion characteristics of people within the monitored area [2,3]. This helps alert staff to observe relevant areas, support the detection of hazardous events, and gather evidence [4]. Currently, CMA technology primarily focuses on the processing of video data [5,6,7]. The related methods can be categorized into two main types: pattern recognition as well as machine learning [8,9], and physical heuristic methods [10].

The first category of methods leverages the currently popular neural networks and machine learning techniques to train or process videos for detecting abnormal behavior. Direkoglu [11] used the amplitude and angle differences of optical flow vectors between consecutive frames to generate a Motion Information Image (MII). This MII is then input into a Convolutional Neural Network (CNN) for training to achieve abnormal classification. Cai et al. proposed the Appearance-Motion Memory Consistent Network (AMMC-Net) [12], which utilizes appearance and optical flow, to increase the differences between normal and abnormal parts in videos. The commonality of the above methods is identifying abnormal frames in videos, but they do not possess the capability of recognizing the types of anomalies or locating abnormal areas. Ganokratanaa et al. introduced a novel unsupervised anomaly detection and localization deep spatiotemporal translation network (DSTN) based on Generative Adversarial Networks (GANs) as well as Edge Wrapping (EW) [13]. This network uses frames of normal events to generate corresponding dense optical flow, serving as temporal features to detect abnormal objects. Colque proposed a spatiotemporal feature descriptor based on optical flow direction, amplitude, and entropy histograms to detect main targets in abnormal events after training [14]. Both Ganokratanaa’s and Colque’s methods were applied for detecting abnormal individuals within crowds. Zhang et al. initially used YOLOv3 for pedestrian detection [15]. Then, they combined a Kalman filter and the Hungarian algorithm to associate detection boxes for multiobject tracking, obtaining the movement trajectories of each pedestrian. Finally, they employed spatial position relationship features to analyze the motion tracking of pedestrians, determining whether the crowd is aggregating or dispersing. In addition, Chondro et al. proposed a motion-based abnormal crowd flow detection method. The Object Vector (OV) was obtained by measuring the weighted time difference between adjacent frames according to spatial mean-sigma observations. It could identify abnormal crowd behavior based on the statistical characteristics of the OV [16]. Wang et al. constructed a large-scale congested crowd counting and localization dataset, NWPU-Crowd [17]. Different from the above methods, Solmaz [18], Wu [19], and Xu [20], proposed some physics-inspired methods. Berkan Solmaz et al. employed linear dynamic systems and Jacobian matrices defined by optical flow to analyze crowd flow in videos, achieving the detection of five collective behaviors: bottlenecks, sources, lanes, arches, and blockages. Shuang Wu et al. introduced a descriptor based on curl and divergence to describe the five motions defined by Solmaz. In addition, this descriptor has the advantages of scale- and rotation-invariance. Xu et al. proposed a comprehensive descriptor by extracting crowd trajectory information, which could efficiently calculate crowd density. Afonso et al. [21] improved target omission in trajectory prediction by employing multiple motion fields.

However, there are still many limitations in practical applications of CMA based on video surveillance. The above methods rely heavily on imaging quality. When lighting conditions are poor or affected by smoke and shadows, the sensitivity of the algorithm is significantly affected. In addition, for video surveillance, targets outside the effective imaging range of the lens (called depth of field) are not clear, which also affects the effect of abnormality detection. Moreover, when moving targets at different distances are projected onto the screen, there are significant differences in the number of pixels occupied by the targets and their degree of motion. In view of this situation, the optical flow field that is obtained directly by using the optical flow method cannot accurately reflect crowd activities. Furthermore, for an image with width *w* and height *h*, the time complexity required to calculate optical flow is roughly O(w×h). As the image resolution increases, the calculation time increases linearly, and it is difficult to ensure real-time performance.

In recent years, the performance of millimeter-wave (mm-Wave) radar has been gradually improved. It possesses advantages such as being unaffected by lighting conditions and capable of operating around the clock, overcoming the drawbacks associated with optical sensors. Some researchers have started using radar to study similar problems [22,23,24,25]. Furthermore, in the discrimination of crowd behavior, mm-Wave radar could directly obtain the position and velocity of targets [26]. Therefore, this paper proposes a method for CMA based on mm-Wave radar point set data, aiming at the detection of abnormal crowd movement. The contributions of this paper are as follows:mm-Wave radar was employed to gather crowd movement information, introducing the concept of RPF. Spatial consistency constraints were enhanced through the fusion of multiple frames, followed by the transformation of optical flow to a binary image by ultilizing neighborhood-based Gaussian smoothing.Based on particle flow information, we derived the virtual motion field of target movement and obtained the particle flow diffusion potential (PFDP). Crowd movement diffusion and aggregation were determined by analyzing extremum values and the spatial distribution of the PFDP.A testing platform was established by using a Texas Instruments (TI) mm-Wave radar to collect continuous frame data under various motion patterns, such as free movement, aggregation, and dispersion. By processing over 20,000 frames of radar raw data, the functionality of the proposed algorithm in crowd behavior discrimination was validated, and the algorithm’s performance was assessed.

The rest of the paper is organized as follows: Section 2 introduces the background knowledge on optical flow methods. Section 3 presents RPF and PFDP, providing a detailed explanation of the proposed method. The experimental validation and algorithm evaluation are described in Section 4. Section 5 concludes the paper.

## 2. Preliminary Knowledge

Optical flow is a computational method used to estimate pixel motion in an image sequence. It is based on the change in pixel brightness in images, calculating the motion vectors of pixels by comparing adjacent frames. With optical flow, the detection of moving targets and estimation of target velocities can be achieved by utilizing only consecutive frames. It is widely applied in various fields of motion analysis as it imposes minimal requirements on the nonmotion attributes of targets. Due to e optical flow being a crucial concept in both images and the method proposed in this paper, it is necessary to introduce optical flow before describing the proposed method.

### 2.1. Optical Flow

Optical flow refers to the instantaneous velocity of spatially moving objects on the imaging plane. It was first proposed by Horn and Schunck in 1981 and was commonly known as the Horn–Schunck (HS) method [27]. It has been widely applied in research areas such as machine-vision-based CMA and anomaly detection. The HS method is based on two fundamental assumptions: brightness constancy and slow motion.

Brightness constancy assumption: This assumption asserts that the brightness of an object remains constant as it moves between frames. It is the most fundamental assumption of optical flow, and various derivative algorithms in optical flow are built upon this premise. The mathematical expression of this assumption is as follows:
(1)I(x,y,t)=I(x+δx,y+δy,t+δt)
where *I* represents the grayscale values at various points in a frame, and it can be viewed as a function of pixel coordinates as well as time (*t*). In most application scenarios, RGB images are typically preconverted to grayscale. This preprocessing step is undertaken to eliminate redundant information, thereby enhancing computational efficiency. Perform Taylor expansion of Equation (Equation 1) at *x* and *y* to obtain
(2)$$\begin{array}{c} I\left(x,y,t\right)=I\left(x,y,t\right)+\delta x\frac{\partial I}{\partial x}+\delta y\frac{\partial I}{\partial y}+\delta t\frac{\partial I}{\partial t}+\varepsilon \end{array}$$
where ε is a second- and higher-order term that includes δx, δy, and δt. Subtract I(x,y,t) from both sides, and then divide by δt to obtain
(3)$$\frac{\delta x}{\delta t}\frac{\partial I}{\partial x}+\frac{\delta y}{\delta t}\frac{\partial I}{\partial y}+\frac{\partial I}{\partial t}+o\left(\delta t\right)=0$$Bring $u=\frac{dx}{dt}$, $v=\frac{dy}{dt}$ into Equation (Equation 3):
(4)$$I_{x}u+I_{y}v+I_{t}=0$$Equation (Equation 4) is called the optical flow constraint equation, which reflects the corresponding relationship between grayscale and velocity.The slow motion assumption: It is assumed that time is continuous, and the motion is slow. The position of the target does not undergo drastic changes over time. It should be noted that the notion of drastic displacement is not easily quantifiable. The trajectory of the target between adjacent frames can be ’approximated’ as continuous in pixels. When the video frame rate is higher or the object’s motion is sufficiently slow, this assumption is easily satisfied. As reflected in Equation (Equation 4), the changes in *u* and *v* as pixels move are slow, and the local region’s variation is small. Especially when the target undergoes nondeformable rigid body motion, the spatial change rate of gthevelocity in the local region is zero. Therefore, a smoothing term is introduced:
(5)$$\begin{array}{c} \zeta_{c}^{2}={\left(\frac{\partial u}{\partial x}\right)}^{2}+{\left(\frac{\partial u}{\partial y}\right)}^{2}+{\left(\frac{\partial v}{\partial x}\right)}^{2}+{\left(\frac{\partial v}{\partial y}\right)}^{2} \end{array}$$For all pixels, the requirement is to solve for *u* and *v* when $\zeta_{c}^{2}$ is at its minimum. Comprehensive optical flow constraints Equation (Equation 4) and velocity smoothing constraints Equation (Equation 5) establish the following energy function:
(6)$$\zeta^{2}=\;\int \int \;\alpha^{2}\zeta_{c}^{2}+{\left(I_{x}u+I_{y}v+I_{t}\right)}^{2}dxdy$$
where α is the smoothing weight coefficient, which represents the weight of the velocity smoothing term. This is a functional extreme value problem, which could be solved using Euler–Lagrange equations. The Euler–Lagrange equations for the first-order partial derivatives of the bivariate function corresponding to Equation (Equation 6) are
(7)$$\left\{\begin{array}{c} \frac{\partial L}{\partial u}-\frac{\partial }{\partial x}\frac{\partial L}{\partial u_{x}}-\frac{\partial }{\partial y}\frac{\partial L}{\partial u_{y}}=0\\ \frac{\partial L}{\partial v}-\frac{\partial }{\partial x}\frac{\partial L}{\partial v_{x}}-\frac{\partial }{\partial y}\frac{\partial L}{\partial v_{y}}=0 \end{array}\right.$$
where
(8)$$\begin{array}{c} L={\left(I_{x}u+I_{y}v+I_{t}\right)}^{2}+\alpha^{2}\left[{\left(\frac{\partial u}{\partial x}\right)}^{2}+{\left(\frac{\partial u}{\partial y}\right)}^{2}+{\left(\frac{\partial v}{\partial x}\right)}^{2}+{\left(\frac{\partial v}{\partial y}\right)}^{2}\right]\; \end{array}$$Bring *L* into Equation (Equation 7):
(9)$$\left\{\begin{array}{c} I_{x}\left(I_{x}u+I_{y}v+I_{t}\right)-\alpha^{2}\nabla^{2}u=0\\ I_{y}\left(I_{x}u+I_{y}v+I_{t}\right)-\alpha^{2}\nabla^{2}v=0 \end{array}\right.$$In [27], the Laplacian operator is approximated by taking the difference between the velocity of a point and the average velocity of its surroundings. That is,
(10)$$\left\{\begin{array}{c} \nabla^{2}u\left(x,y\right)=u^{n+1}\left(x,y\right)-{\bar{u}}^{n}\left(x,y\right)\\ \nabla^{2}v\left(x,y\right)=v^{n+1}\left(x,y\right)-{\bar{v}}^{n}\left(x,y\right) \end{array}\right.$$Substitute Equation (Equation 10) into Equation (Equation 9) to obtain the iteration relationship:
(11)$$\left\{\begin{array}{c} \left(I_{x}^{2}+\alpha^{2}\right)u^{n+1}+I_{x}I_{y}v^{n+1}=\alpha^{2}{\bar{u}}^{n}-I_{x}I_{t}\\ \left(I_{y}^{2}+\alpha^{2}\right)v^{n+1}+I_{x}I_{y}u^{n+1}=\alpha^{2}{\bar{v}}^{n}-I_{y}I_{t} \end{array}\right.$$By simplifying Equation (Equation 11), iterative formulas for u and v are obtained using Equation (Equation 12)
(12)$$\left\{\begin{array}{c} u^{n+1}={\bar{u}}^{n}-\frac{I_{x}\left(I_{x}{\bar{u}}^{n}+I_{y}{\bar{v}}^{n}+I_{t}\right)}{\alpha^{2}+I_{x}^{2}+I_{y}^{2}}\\ v^{n+1}={\bar{v}}^{n}-\frac{I_{y}\left(I_{x}{\bar{u}}^{n}+I_{y}{\bar{v}}^{n}+I_{t}\right)}{\alpha^{2}+I_{x}^{2}+I_{y}^{2}} \end{array}\right.$$

### 2.2. Lucas–Kanade (LK) Method and Farneback Method

In order to further improve the accuracy of optical flow estimation when the target moves quickly, the Lucas–Kanade method enhances the HS method. Unlike the HS method, the LK method [28] assumes that the local brightness in an image is constant, meaning that within the same local area, brightness changes satisfy Equation (Equation 4). Additionally, it introduces a spatial consistency constraint. Substituting the points to be determined, along with their neighborhood points, into Equation (Equation 4) produces
(13)$$II\left[\begin{array}{c} u\\ v \end{array}\right]=\left[\begin{array}{c} -I_{t1}\\ -I_{t2}\\ \vdots \\ -I_{t9} \end{array}\right]$$
where:(14)$$II=\left[\begin{array}{cc} I_{x1} & I_{y1}\\ I_{x2} & I_{y2}\\ \vdots \\ I_{x9} & I_{y9} \end{array}\right]$$

Both sides of Equation (Equation 13) are simultaneously multiplied by $II^{T}$
(15)$$\left[\begin{array}{cc} \Sigma I_{x}^{2} & \Sigma I_{x}I_{y}\\ \Sigma I_{x}I_{y} & \Sigma I_{y}^{2} \end{array}\right]\left[\begin{array}{c} u\\ v \end{array}\right]=-\left[\begin{array}{c} \sum I_{x}I_{t}\\ \sum I_{y}I_{t} \end{array}\right]$$*u* and *v* in Equation (Equation 15) can be solved by using the least squares method.

Furthermore, Farneback [29] utilized a quadratic polynomial to model local signals:(16)$$f\left(x\right)∼x^{T}Ax+b^{T}x+c$$
where $x={\left[x,y\right]}^{T}$, **A** is a symmetric matrix, **b** is a vector, and *c* is a scalar. The polynomial coefficients are estimated through a weighted least squares fit to the values of the neighborhood signals. The objective function is established based on the gradient invariance of the local brightness:(17)$$e\left(x\right)=\sum_{\Delta x\in I}w\left(\Delta x\right){\left\|A\left(x+\Delta x\right)d-\Delta b\left(x+\Delta x\right)\right\|}^{2}$$
where d=∂x/∂t, representing the pixel displacement per unit time, is equivalent to *u* and *v* in the HS or LK methods. In the sections below, d is consistently denoted as ($v_{x}$, $v_{y}$). *w* is the weight of the neighborhood for *x*, which is assigned using a Gaussian filter.

In the actual application of the Farneback algorithm, in order to handle larger motion, an image pyramid structure is introduced. This is achieved by downsampling the image, reducing the magnitude of the target’s motion. The commonly used downsampling calculation method is
(18)$$\begin{array}{cc} I^{L}\left(x,y\right)= & \frac{1}{4}I^{L-1}\left(2x,2y\right)+\frac{1}{8}[I^{L-1}\left(2x-1,2y\right)+I^{L-1}\left(2x+1,2y\right)+\\ & I^{L-1}\left(2x,2y-1\right)+I^{L-1}\left(2x,2y+1\right)]+\frac{1}{16}[I^{L-1}\left(2x-1,2y-1\right)+\\ & I^{L-1}\left(2x+1,2y+1\right)+I^{L-1}\left(2x-1,2y+1\right)+I^{L-1}\left(2x+1,2y+1\right)] \end{array}$$
where *L* is the current level of the image pyramid.

### 2.3. The Computational Complexity of the Optical Flow Method

According to Equation (Equation 12), we notice that $v_{x}$ and $v_{y}$ require initial values and undergo a finite number of iterations to obtain the final result. The number of iterations directly affects the accuracy of the optical flow calculation. Another major factor consuming computational resources in this method is the calculation of gradients and means, where the time complexity is O(w×h), with *w* being the number of columns in the image matrix and *h* being the number of rows. If a total of *n* iterations are set, then the complexity of running this method once to obtain optical flow between two frames is O(w×h×n). The effects and runtime of the above three optical flow algorithms are presented in the fourth section of this article. But, it is feasible to analyze crowd movements by treating the crowd as a collection of particles. From this perspective, the information provided by optical imaging is predominantly redundant. As radar does not require attention to imaging details, the data volume is relatively small. Additionally, the results calculated in radar plane coordinates can provide accurate distance and azimuth information. With the advantages of all-day and all-weather operation, mm-Wave radar was employed in this study for this problem.

## 3. Methodology

This section introduces radar particle flow (RPF) and particle flow diffusion potential (PFDP), which are integral components of the research methodology. PFDP is employed to estimate the distribution of diffusion sources and subsequently localize dispersion centers. Figure 1 illustrates the fundamental structure of the CMA system based on mm-Wave radar.

The workflow proceeds as follows:(1)The raw Analog-to-Digital Converter (ADC) data used to capture crowd echoes can be transformed into 3D Fast Fourier Transform (FFT) to obtain distance and angle information of measurement points, thereby exporting a mm-Wave radar point set.(2)Coordinate transformation is applied to project all measurement points onto pixel coordinates, forming a pixel map.(3)The optical flow field is computed between adjacent frames, and RPF is obtained through Gaussian smoothing and local maxima sampling.(4)Utilizing the direction and position information of vectors at various points in the flow field, the virtual distribution of sources at diverse locations in the plane is inferred, and a new concept of micro-source is defined.(5)The micro-sources are integrated within a closed rectangular window, iterating over the entire RPF plane to generate a matrix, which describes the potential distribution at various locations in the plane.(6)Over a certain period, these potential matrices are superimposed to form a potential surface, delineating aggregation and divergence points while effectively characterizing the flow direction of the crowd.

We describe the entire method in three parts: Section 3.1 describes the process of generating a binary pixel map from the crowd radar echoes. Section 3.2 details the method of obtaining RPF by using optical flow and adjacent-frame mm-Wave point sets. Section 3.3 builds upon Section 3.2, proposing the micro-source (MS) and PFDP for CMA.

### 3.1. Acquisition and Preprocessing of Point Set Pixel Images

The operation of single-chip mm-Wave radar is based on the principles of frequency-modulated continuous wave (FMCW), enabling it to measure the range, relative radial speed, and angle of targets. Figure 2 shows the process of an FMCW radar collecting echoes from crowd movements and handling them into point sets. The position relationship between the radar and the object in the coordinate system is shown in Figure 3. The FMCW radar system emits continuous signals to the target audience through the transmitter (TX) antenna, which are then reflected and captured by the receiver (RX) antenna. The received signal is mixed with the transmitted signal to generate an intermediate frequency (IF) signal. The signal is further processed through a bandpass filter to eliminate high-frequency noise and converted into a digital signal by using an ADC. This digitized signal then undergoes various digital signal processing (DSP) procedures to derive the raw radar data. By performing a 3D-FFT (Range-FFT, Angle-FFT, Doppler-FFT) analysis on the raw radar data, the distance $r_{i}$, radial velocity $v_{i}$, and azimuth angle $\theta_{i}$ information of the echoes can be obtained. In each frame, the distance information corresponding to the *i*th echo signal is given by
(19)$$r_{i}=\frac{\phi_{0}\lambda }{4\pi }$$

Here, $\phi_{0}$ is the phase obtained from the IF signal $A\sin\left(2\pi f_{0}t+\phi_{0}\right)$, and λ represents the wavelength.
(20)$$v_{i}=\frac{\lambda \Delta \phi }{4\pi T_{c}}\;,\;\left(v_{i}<\frac{\lambda }{4T_{c}}\right)$$

To measure speed, the FMCW radar emits two chirps with an interval of $T_{c}$. When each returned chirp reaches a peak at the same position over the distance FFT, their phase difference is Δϕ. The azimuth angle corresponding to the *i*th echo is:(21)$$\theta_{i}=\sin^{-1}\left(\frac{\lambda \Delta \phi }{2\pi l}\right)$$

Here, *l* represents the distance between the two receiver (RX) antennas. Δϕ denotes the phase difference in the echo signal of the same distance FFT peak received between two different chirp cycles.

For ease of the subsequent expressions, we consistently represent target points in Cartesian coordinates as positions ($x_{i}$, $y_{i}$). The upper part of Figure 4 shows the radar point cloud data in scenarios of free walking and rapid departure. The lower part corresponds to the motion videos.

As ($x_{i}$, $y_{i}$) is based on the coordinate system with the radar’s position as the origin, the image used for optical flow calculation often takes the top-left pixel as the coordinate origin. Therefore, in order to estimate target motion vectors by using optical flow, the radar data need to undergo the following transformation to pixel coordinates. To achieve this, firstly, create a pixel matrix $I_{R}\left(m\times n\right)$ with dimensions m×n, initializing its internal elements to zero.
(22)$$I_{R}\left(m,n\right)=0$$

Then, using the radar target’s position coordinates as indices, set the corresponding elements in $I_{R}$ to 1.
(23)$$I_{R}\left(\left[m-y_{i}\right],\left[x_{i}+\;o\right]\right)=1$$

As radar coordinates are floating-point numbers while image coordinates are integers, rounding [.] is necessary during the conversion from radar coordinates to pixel coordinates. *o* is a horizontal offset greater than 0. Its purpose is to map all point targets in the scene to the matrix index range. In Equation (Equation 23), a binary image generated from a single-frame radar data is obtained. The selection of *m* and *n* depends on the scene size, radar distance resolution, and angle resolution. In our experiments, considering both hardware parameters and target dimensions, a unit pixel interval of 0.1 m was chosen, and both width and height were set to 300 units.

### 3.2. Radar Particle Flow

In this subsection, the radar particle flow is defined as a vector field, which is composed of the instantaneous velocities for all point set particles in the plane at the current moment. This vector field can be obtained by calculating the optical flow of the point set pixel video. Therefore, the assumptions and methods for computing optical flow should also be satisfied when calculating the RPF. The sources and potential distribution of this vector field can reflect the activities of the crowd. RPF can be described as
(24)$$\left.RPF=\left\{\begin{array}{c} v_{x}\left(r,c\right)\\ v_{y}\left(r,c\right) \end{array}\right.,\left(1\leq r\leq m,1\leq c\leq n\right)\right.$$

Due to the phenomenon of target loss in the process of radar target detection, it is challenging to ensure that all targets can be detected in every frame. When a target is lost, the assumption of constant brightness does not hold, leading to suboptimal results in directly calculating the RPF. Therefore, we optimized the calculation of RPF from three aspects:Merging data across multiple frames can help suppress target disappearance. For example, merging every 5 original frames into 1 combined frame and then calculating the RPF based on adjacent pairs of these combined frames, as shown in Figure 5.To enhance the constraint of spatial consistency, it was necessary to expand the neighborhood of optical flow calculation. In this study, since the objects were mm-Wave radar point clouds with no fixed shape, an adaptive method was needed to determine the neighborhood size (NS). The process in this method is as follows:Input a set of combined frame mm-Wave point set data: $X=\{x_{1},x_{2},\ldots ,x_{n}\}$. Since the point set of the targets within the combined frame exhibits a certain degree of extension in the direction of travel, similar to an ellipse, we chose the adaptive ellipse distance density peak fuzzy (AEDDPF) [30] method to cluster $x_{i}\in R^{2}$ and obtain *K* clusters.For each cluster $C_{k}$, calculate the geometric center:
(25)$$c_{k}=\frac{1}{|C_{k}|}\sum_{x_{i}\in C_{k}}x_{i}$$
where $|C_{k}|$ represents the number of points belonging to the *k*th cluster.Take the geometric center $c_{k}$ of each cluster $C_{k}$ as the center to construct a circle:
(26)$$C_{k}=\left\{\left(x_{i},r_{k}\right)|x_{i}\in C_{k}\right\}$$
where:
(27)$$r_{k}=\max_{x_{i}\in C_{k}}(∥c_{k}-x_{i}∥)$$Here ‖.‖ represents distance.Finally, calculate the average diameter of all clusters as NS:
(28)$$NS=\frac{1}{K}\sum_{i=1}^{K}d_{i},d_{i}=2\cdot r_{i}$$Next, run the Farneback algorithm on consecutive frames to compute the RPF based on mm-Wave point set images.In order to obtain the overall velocity direction of targets, it is necessary to downsample the RPF of the target and its neighboring pixels. Since the shape of the radar point set for a target is unstable, unlike optical imaging, RPF cannot utilize feature point tracking [31,32] to obtain the target’s velocity. Considering that the velocity magnitudes of the target point and various pixel points in its neighborhood follow a Gaussian distribution, the direction corresponding to the maximum velocity magnitude within that neighborhood can be determined as the overall velocity direction of the target. That is,
(29)$$\left\{\begin{array}{c} RPF\left(x_{i},y_{i}\right)=Max\left\{M\left(x,y,\theta \right)|\left(x,y\right)\in \epsilon \left(x_{i},y_{i}\right)\right\}\\ M\left(x,y\right)=\sqrt{{v_{x}}^{2}+{v_{y}}^{2}}\\ \theta \left(x,y\right)=arctan\frac{v_{y}}{v_{x}} \end{array}\right.$$
where $\delta (x_{i},y_{i})$ is the ϵ neighborhood centered at point $\left(x_{i},y_{i}\right)$, and its size is generally about NS.

Figure 6 illustrates the process of local maximum sampling for the RPF. We conducted RPF extraction on a motion scene composed of five individuals, which is shown in Figure 6a. Initially, we used the mm-Wave radar to collect data from the scene and performed the initial RPF extraction. Figure 6b displays the extracted targets and their corresponding velocity vectors in the neighborhood, forming the initial RPF. In Figure 6c, the maximum values of the velocities in Figure 6b are visually evident. Then, by extracting the velocity vectors corresponding to these maximum values, the velocity directions for each target is obtained, as shown in Figure 6d.

### 3.3. Micro-Source (MS) and PFDP

The crowd’s movement, which is converted into particle group dynamics in the pixel plane in Section 3.1, underwent further processing in Section 3.2. This processing resulted in the computation of the velocity vector field for the particle group, referred to as the RPF. The crowd’s movement could be considered as particles responding to the field, which would converge to or diverge from the source. In order to visually display the field and source, the potential energy diagrams of two typical crowd movements are presented in Figure 7. From the graph, it can be seen that the maximum or minimum points of potential energy can be identified, whether it is aggregation or diffusion. Therefore, it is feasible to perform anomaly detection and locate anomaly centers based on the potential RPF energy.

A real source field can use the field vector as a gradient to inversely solve the potential energy surface through a double integration on the plane or infer the size and position of the source using Gauss’s theorem. However, the RPF is not a physical field that exists in reality; it only generates values where there is target movement. Referring to the calculated results in Figure 6, the radar particle flow field is usually incomplete across the entire detection range. Although there are many studies [33] on reconstructing the flow field based on incomplete data, they are not applicable to this study. Therefore, we developed a new concept called “micro-source (MS)” and then estimated the position and potential energy distribution of the source in the RPF through MS estimation.

**Definition** **1.**
*Under the assumption of linear motion within Δt, MS, denoted as $\delta_{i,j}$, is a vector that is used to describe whether any two RPF vectors originate from or reach the same location:*

(30)
$$\delta \left(i,j\right)=\left[g_{i,j},x_{i,j},y_{i,j}\right],\left(i\neq j\right)$$



$\delta_{i,j}$ consists of two parts, where $g_{i,j}$ is used to describe whether the *i*th and *j*th RPF vectors come from or flow to the same position, and $\left(x_{i,j},y_{i,j}\right)$ is used to describe the specific coordinates of yhe MS. After obtaining the velocity vectors $\vec{\alpha_{i}}=\left(v_{xi},v_{yi}\right)$ and $\vec{\alpha_{j}}=\left(v_{xj},v_{yj}\right)$ of the two target points in the RPF, according to the intersection relationship of their respective straight lines, there are four cases for $g_{i,j}$:(a)These two vectors originate from the intersection point, and, at that point, $g_{i,j}$ = 1.(b)Both vectors point toward the intersection point, and, at that point, $g_{i,j}$ = −1.(c)One vector points to the intersection point, and the other vector points away from the intersection point; at that point, $g_{i,j}$ = 0.(d)In other cases, $g_{i,j}$ is undefined.

Based on the above description, $\delta_{i,j}$ could mark the position of the MS for any pair of moving particles. In addition, it can be analyzed for the net flow at any point in the RPF. At the position where $g_{i,j}\neq 0$, the point has weak divergence or convergence characteristics, similar to the physical meaning of divergence. In surface integration, $g_{i,j}=+1$ represents that the associated vector points from the inside to the outside of the integration surface, and vice versa.

The specific steps for solving δ are as follows:

Firstly, calculate the intersection coordinates based on the two vectors $\vec{\alpha_{i}}$ and $\vec{\alpha_{j}}$ extracted from the RPF. Establish a system of equations based on the corresponding straight lines of these two vectors:(31)$$\left[\begin{array}{cc} k_{i} & -1\\ k_{j} & -1 \end{array}\right]\left[\begin{array}{c} x\\ y \end{array}\right]=\left[\begin{array}{cc} k_{i}x_{i} & y_{i}\\ k_{j}x_{j} & y_{j} \end{array}\right]\left[\begin{array}{c} 1\\ -1 \end{array}\right]$$
where $\left(x_{i},y_{i}\right)$ and $\left(x_{j},y_{j}\right)$ represent the target points of the two vectors. The slopes are defined as $k_{i}=v_{yi}/v_{xi}$ and $k_{j}=v_{yj}/v_{xj}$. Solve the intersection coordinates:(32)$$\left[\begin{array}{c} x\\ y \end{array}\right]={\left[\begin{array}{cc} k_{i} & -1\\ k_{j} & -1 \end{array}\right]}^{-1}\left[\begin{array}{cc} k_{i}x_{i} & y_{i}\\ k_{j}x_{j} & y_{j} \end{array}\right]\left[\begin{array}{c} 1\\ -1 \end{array}\right]$$

Then, to solve $g_{i,j}$ at the intersections, we construct two auxiliary vectors:(33)$$\vec{\beta_{i}}=\left(x_{i}-x,y_{i}-y\right),\vec{\beta_{j}}=\left(x_{j}-x,y_{j}-y\right)$$

And, define the discriminant as follows:(34)$$\Delta_{i}=sign\left(\vec{\alpha_{i}}\cdot \vec{\beta_{i}}\right),\;\Delta_{j}=sign\left(\vec{\alpha_{j}}\cdot \vec{\beta_{j}}\right)$$

The signs of $\Delta_{i}$ and $\Delta_{j}$ reflect the position relationship of the intersection point (x,y) with respect to $\vec{\alpha_{i}}$ and $\vec{\alpha_{j}}$, respectively. Referring to Figure 8, calculate the value of $g_{i,j}$ using Equation (Equation 35).
(35)$$g_{i,j}=\frac{1}{2}\left(\Delta_{i}+\Delta_{j}\right)$$

Then, following the method described in Equation (Equation 31) to (Equation 35), iterate through all RPF vectors to obtain all $g_{i,j}$ of the RPF. Establish a mapping function based on the position of all obtained $g_{i,j}$:(36)$$f\left(x,y\right)=\left\{\begin{array}{c} g_{ij}\left(x,y\right),\vec{\alpha_{i}}\times \vec{\alpha_{j}}\neq 0\\ 0,other \end{array}\right.$$

After that, apply a double integral to f(x,y) in the specified region *s*, resulting in the particle flow diffusion potential $\left(PFDP_{s}\right)$ within region *s*.
(37)$$PFDP_{s}={\;\int \int \;}_{s}f\left(x,y\right)dxdy\Leftrightarrow \oint \vec{RPF}\cdot ds,s\subseteq S_{RPF}$$

$PFDP_{s}$ describes the ability of a specified region to disperse global particles. The larger the absolute value of $PFDP_{s}$, the denser same-signed $g_{ij}$ values are within the specified *s*, indicating a stronger dispersive capability of *s* on global particles.

In practical implementation, considering that pixel coordinates are integers, each position attribute of δ is rounded to the nearest integer based on the nearest-neighbor rule as an index. The $g_{ij}$ value of δ is accumulated in the index, resulting in a global distribution matrix, which is denoted as distribution (*D*). Subsequently, by setting *s* as a sliding window, convolution filtering is applied to the entire *D* to calculate the $PFDP_{s}$ at each location, obtaining the global PFDP matrix. A Gaussian kernel is employed in this convolution, which allows the computed diffusion sources to be more concentrated, enhancing the distinctiveness from source-free regions and ensuring smooth variations.
(38)$$PFDP\left(r,c\right)=D\left(r+p,c+q\right)*\left(\frac{1}{2\pi \sigma^{2}}\cdot e^{-\frac{p^{2}+q^{2}}{2\sigma^{2}}}\right)$$
where *p* and *q* are independent variables of two-dimensional Gaussian functions. *r* and *c* run over all values to produce the global PFDP matrix. The entire process from inputting the RPF to obtaining the PFDP matrix is described in Algorithm 1.

Through Algorithm 1, three important parameters are recorded:PFDP Matrix: Obtained from Equation (Equation 38), it describes the distribution of PFDP intensity within a single frame. The feature energy map generated using the PFDP matrix intuitively reflects the strength of the PFDP at various pixel coordinates at the current moment.The maximum absolute value of elements in PFDP per frame (MAVEPF): The MAVEPF curve effectively characterizes the variation in the state of crowd movement. It is important to note that due to the influence of different data sources, the extracted amount of RPF varies. Therefore, we do not define a fixed threshold for distinguishing anomalies in the MAVEPF curve. The MAVEPF curve can be used to observe the relative differences between frames in the same data source.Accumulated PFDP: It is obtained by accumulating continuous frames of the PFDP and can be presented in the form of a three-dimensional surface. The accumulation of the PFDP helps suppress the influence of random noise and balances the bias in event localization. The highest point on the PFDP surface represents the location with the strongest dispersion capacity in the current environment, while the lowest point indicates the location with the strongest absorption capacity. These points are considered the central locations most likely to experience diffusion or aggregation phenomena.
**Algorithm 1:** RPF to PFDP based on MS
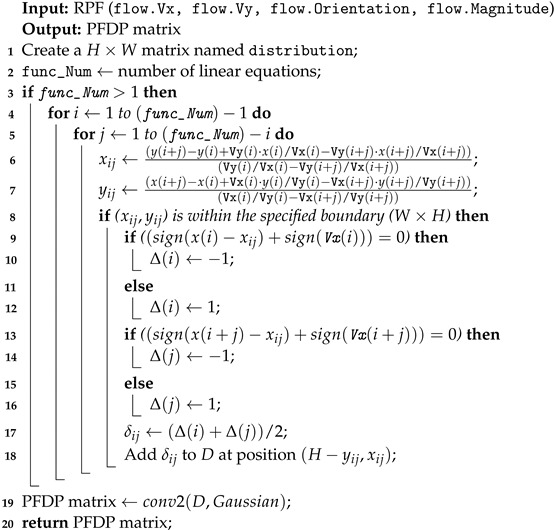


## 4. Experiments and Analysis

In order to verify the functionality and test the performance of the proposed method, we conducted relevant experiments. Firstly, the experimental setup is described, including an introduction to the software and hardware platforms, as well as the data collection process. Subsequently, the collected data processing is explained for different scenarios such as normal crowd movement, sudden fleeing, and gathering. Finally, the performance of the proposed method is compared with that of classical video-based methods. The processed results validate the effectiveness of the proposal, and the performance of the proposed method was evaluated.

### 4.1. Experimental Setup


This study used the IWR1642BOOST mm-Wave radar evaluation board from Texas Instruments Incorporated (Dallas, Texas, USA) for experimental purposes. IWR1642BOOST is a single-chip frequency-modulated continuous-wave (FMCW) radar that integrates a PLL, a transmitter, a receiver, an ADC, and an on-chip digital signal processing (DSP) unit. The radar was equipped with four receiving channels and two transmitting channels, having an output power of up to 12.5 dBm. Operating in the frequency range of 76 GHz to 81 GHz, it offered a continuous usable bandwidth of 4 GHz. Table 1 provides detailed specifications of this radar. Raw data from the radar were saved in binary format (.bin) to a computer by using the DCA1000EVM data capture board. Processing these binary files allowed for the extraction of point set information such as distance, azimuth, and radial velocity of multiple targets. Figure 9 depicts the radar during outdoor data collection. Additionally, the hardware we used in this study included an AMD Ryzen 5 3500x6-core 3.6 GHz CPU and 8 GB RAM. All calculated graphs were generated by using Matlab R2021b.

### 4.2. Sudden Diffusion and Aggregation

To validate the proposed algorithm’s ability to detect abnormal crowd behavior and locate potential anomalous centers, experiments were conducted in two outdoor scenarios. In Scene 1, crowd movements in a circular square were monitored, with the radar positioned 14 m directly in front of the center of the square, as shown in Figure 10. Scene 2 involved observing crowd movements on an athletic track, as shown in Figure 11. A cross mark was placed on the track, and the radar was positioned 9 m in front of the cross mark. During the experiments, people freely moved within the specified areas and upon receiving instructions, simultaneously fled away from the mark or gathered toward it from the outer side. In both Scene 1 and Scene 2, 50 sets of crowd movements with fixed points for dispersal and 50 sets for gathering were collected, resulting in a total of 200 sets of data, comprising approximately 20,000 frames.

Figure 12 shows the MAVEPF curves for the four sets of experiments described in Figure 10 and Figure 11, with the red lines indicating the moments when the crowd received instructions. It can be observed that all MAVEPF curves rapidly rise on the right side of the red line, contrasting with the left side. This indicates a rapid increase in the density of the same type of MS within a certain area after the red line moment, which is significantly different from normal motion, indicating a change in the state of crowd movement.

Figure 13 shows the corresponding RPF, downsampled RPF as well as micro-sources for the four sets of experiments, the PFDP of the current frame, followed by the accumulated PFDP surface, which starts from a normal frame until the occurrence of an anomaly. At this point, observe the main peak of the PFDP surface to identify the center of abnormal movement. If the main peak is upward, then the position of the main peak is the evaluated diffusion center; otherwise, it is the aggregation center. In the experiments, the radar was placed at the pixel coordinates (150, 300).

The results of abnormal center localization from a total of 200 experiments are summarized in Figure 14. Among them, the anomaly centers evaluated in each experiment are marked with blue dots. Additionally, we calculated the mean abnormal center (green circle) for each group of 50 experiments and annotated it together with the marked reference points (red star) in the respective figures. The mean absolute error (MAE) between the estimated abnormal center and the reference point for each 50 experiments is recorded in Table 2. Furthermore, Table 2 provides statistics on the discrimination of abnormal types in each scene.
(39)$$\mathrm{MAE}=\frac{1}{N}\sum_{k=1}^{N}\sqrt{\left({\left(x_{k}-x_{RP}\right)}^{2}+{\left(y_{k}-y_{RP}\right)}^{2}\right)}$$
where *N* represents the number of experiments in each group, ($x_{k},y_{k}$) represents the coordinates of the main peak of PFDP in each experiment, and ($x_{RP},y_{RP}$) represents the reference point for each group of experiments.

### 4.3. Normal Pedestrian Flow

To validate that the proposed algorithm does not produce false alarms during normal pedestrian flow, the radar was set up approximately 10 m in front of the entrance to the campus canteen. During peak dining hours, a large number of people approached the canteen from two main directions, walking normally. During this period, the crowd exhibited stable flow. Two long-duration continuous datasets were obtained, totaling 2000 frames. The only difference between them was the varying degree of crowd density. The pedestrian movements in these two experimental groups are shown in Figure 15. The algorithm performance and MAVEPF curves for these two sets of experiments are collectively presented in Figure 16. It can be observed that the MAVEPF curves of the two datasets do not show significant changes throughout the entire process, indicating normal pedestrian movement and no sudden events. A large value was generated at frame 105 in Figure 16a, but the MAVEPF at the same level did not continue to appear. It can be considered as an accidental event. In contrast, the algorithm performed more stably in environments with denser crowds, was shown in Figure 16b. Meanwhile, the PFDP surfaces still reflected the flow of people over time, as shown in Figure 17. Moreover, both PFDP surfaces indicated the same gathering point (the location of the canteen entrance). Despite the difference in crowd density between the experiments, the results remained consistent, demonstrating the effectiveness of the PFDP surface in reflecting the movement status of the crowd.

### 4.4. A Random Scene of Roller Skating Training Class

This was a sports activity clip captured randomly in the Olympic Park. In this scene, the training of roller-skating students was recorded. Initially, about 20 students gathered around the coach’s location, remained stationary for a few seconds, then moved counterclockwise around the area at different radii, and finally regrouped around the coach before coming to a halt. Throughout this period, the coach remained stationary. This process was stored as 1000 frames of raw radar data. Figure 20 shows images and point sets at four moments in these data. The students gathered for the first time from t = 0 to t = 3, as shown in (a) and (b), which corresponds to frames 0 to 15 of radar data. The second time was from t = 30 to t = 32, as shown in (c) and (d), corresponding to frames 150 to 160 of radar data. After algorithm processing, it can be observed in Figure 18 that there was a significant, continuous increase in the MAVEPF curve during student aggregation. The red line marks the starting frame of the second aggregation. When the MAVEPF value was 0, the target group was stationary, and the radar did not detect the moving target. In Figure 19, it can be observed that the accumulation of two aggregation processes on the PFDP surface resulted in a main peak pointing downward, which was the position of the coach. We mark the coordinates of the main peak in the first column in Figure 19, and we mark this position in Figure 20. Comparing these three figures shows that the proposed method in this experiment correctly determined clustering and indicated the center of clustering. The proposed algorithm effectively perceived aggregation events in this dataset and located the areas where events occur.

In these experimental scenarios, the proposed algorithm ran on an environment consisting of an AMD Ryzen 5 3500x six-core 3.6 GHz CPU, 8 GB of RAM, and MATLAB. The entire duration of the corresponding events was 100 s. The total runtime of the proposed algorithm was 30.57 s, satisfying the real-time processing requirements.

### 4.5. Algorithm Performance Analysis

In this subsection, the proposed method is compared with traditional video-based methods in terms of extracting crowd motion information and detecting crowd anomalies.

#### 4.5.1. Performance Comparison of Extracting Crowd Movement Information

To verify the ability of the proposed method to reflect crowd movement for RPF, we analyzed video and radar data in the same scene. The resolution of the video was 1280 × 720, with 30 frames per second. The radar collected 10 frames of data per second during operation.

Firstly, traditional methods such as HS, LK, and Farneback were used to process video data in a diffusion scenario and extract the optical flow of crowd motion. Meanwhile, the RPF was extracted using the proposed method in the same scene, which was compared with the optical flow of the three video-based methods. Figure 21 shows the optical flow and RPF results obtained using the four methods. Among them, the HS method and LK method exhibited false motion directions in areas with boundary line features of the image, as shown by the white runway lines in Figure 21a,b. The Farneback method effectively avoided false optical flow in the background. However, due to the fact that the human shadow changes with human movement, the Farneback method exhibited some false optical flow in the area of human shadows, as shown in Figure 21c. The proposed method extracted the RPF based on radar echoes and was not affected by factors such as light and contrast. From Figure 21d, it can be seen that the RPF not only reflected the movement direction of each target but also had minimal background interference. Therefore, the proposed method showed the ability to accurately capture crowd movements. It is evident that the performance and efficiency of the Farneback method are consistent with the conclusions drawn in [34].

On this basis, we statistically analyzed the running time of the four methods mentioned above when extracting optical flow or RPF. From Figure 22, it can be seen that under the same conditions, processing 300 frames of video or radar data, the Farneback method had the longest running time, reaching 109 s. The operation time for extracting the RPF by using the proposed method was similar to that of the HS and LK methods, at only 5.18 s. It can be seen that the proposed method is less computationally complex in the extracting RPF. It can be seen that the proposed method is more computationally efficient in extracting the RPF. The calculation parameters used in the Farneback algorithm are shown in Table 3

#### 4.5.2. Performance Comparison in Detecting Crowd Anomalies

To demonstrate the performance of the proposed algorithm, the proposal based on mm-Wave radar and the classic social force model (SFM) method [35] based on video were applied for aggregation and diffusion behaviors from Scene 1 and Scene 2 in Section 4.2. For the same scene, the proposed algorithm was applied to radar data, while the corresponding video data were processed using the SFM method. Samples illustrating crowd dispersion and aggregation behaviors in Scene 1 are shown in Figure 23 and Figure 24, respectively. Figure 25 and Figure 26 depict samples from Scene 2. In these figures, (a) represents the optical flow with yellow lines and social force flow (SFF) with yellow lines based on the SFM method; (b) shows the RPF of the crowd by using radar data; (c) displays the social force intensity map from the video, where regions with concentrated social force intensity are considered as regions of interest (ROI) by the SFM model; (d) presents the PFDP, representing the ROI determined using the proposed method. In these four sets of images, the left-hand side (a) and (c) was generated using the video-based SFM, while the right-hand side (b) and (d) was generated by using radar-based RPF. From Figure 23, Figure 24, Figure 25 and Figure 26, it can be seen that

In (a) in Figure 23, Figure 24, Figure 25 and Figure 26, the effectiveness of extracting optical flow from video data using the SFM method is demonstrated, as shown by the yellow lines, which represent the flow direction of the targets in the video. In (b), we can see that the proposed method also detected the flow direction of the target from radar data. Compared to the video method, RPF also provided positional and directional information on a two-dimensional plane, thereby offering a more realistic representation of the target’s flow direction.From (c) in Figure 23, Figure 24, Figure 25 and Figure 26, it can be observed that the SFM focused on the intersection area of dense flows. In contrast, the proposed method was more inclined toward detecting the source of crowd movement or the center of arrival. For instance, in Figure 23d and Figure 25d, the yellow region represents the center position where the crowd disperses away during diffusion, while the blue region in Figure 24d and Figure 26d indicates the center position where the crowd gathers during aggregation.In addition, it can be seen from Figure 26a that the SFM method generates false SFF when calculating SFF due to the motion of ground shadows in the video, as shown in Figure 26c. The radar based method can avoid the impact of optical conditions such as lighting and contrast.

Based on the above experiment, we also compared the runtime of the SFM method and the proposed algorithm to reflect the computational complexity of the two algorithms. Ten groups of aggregation samples and ten groups of diffusion samples from each of Scene 1 and Scene 2 were selected to record runtime for a total of 40 sets of data. For the SFM method, each group sample consisted of 100 frames video data, with an image resolution of 1280 × 720, corresponding to an activity duration of 3.3 s. For the proposed method, each group sample included 100 frames radar data, corresponding to an activity duration of 10 s. The average runtime of the two methods under different motion modes in Scene 1 and Scene 2 is shown in Table 4. It can be seen that the proposed method only required about 3 s to process 100 frames radar data, which is much lower than the time cost for SFM to process video data.

## 5. Conclusions

This paper proposed a crowd movement analysis method based on mm-Wave radar. By extracting radar particle flow from mm-Wave point set data, the micro-source distribution in the scene was calculated, and the PFDP at each location was derived. By accumulating the PFDP over a period of time, the PFDP surface was obtained, and the corresponding potential energy extremum point was found to locate the diffusion or aggregation center. The MAVEPF curve characterized the changes in the movement status of a crowd. Over 20000 frames of raw radar data were processed by using the proposed algorithm to verify its effectiveness. The experimental results showed that the proposed algorithm effectively identified behaviors such as aggregation and dispersion, with an accuracy of no less than 88%, and the positioning accuracy of the event source was within 2 m. The proposed algorithm processed 10 frames of radar data per second, which required 0.3 s, meeting real-time computing requirements.

## Figures and Tables

**Figure 1 sensors-24-01899-f001:**
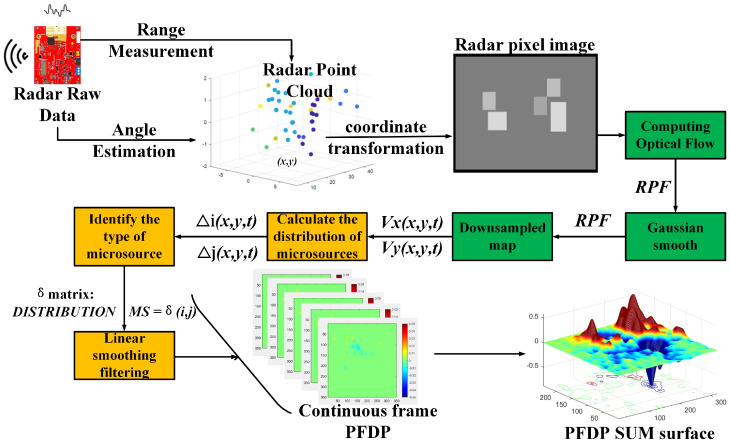
The basic structure of a CMA system based on mm-Wave radar.

**Figure 2 sensors-24-01899-f002:**
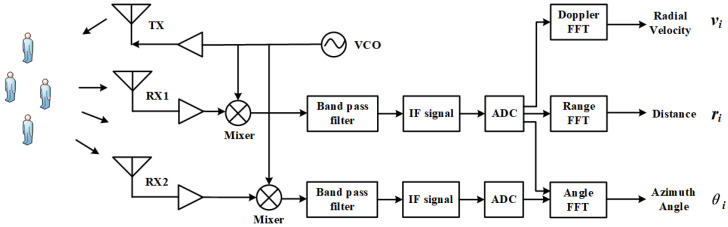
FMCW radar signal processing.

**Figure 3 sensors-24-01899-f003:**
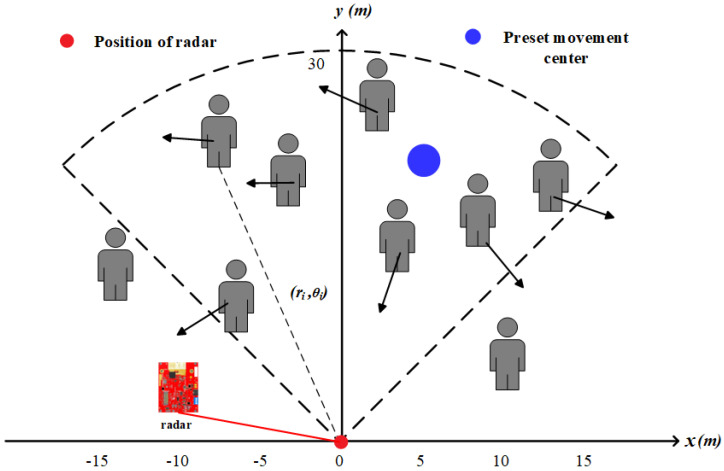
The position relationship between the radar and the measured object.

**Figure 4 sensors-24-01899-f004:**
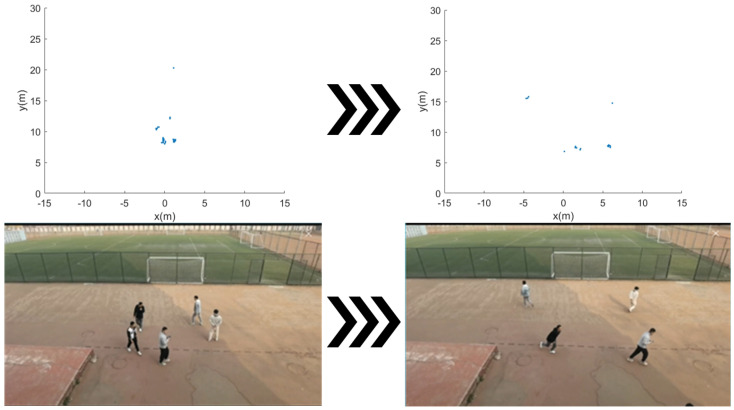
Comparison between the mm-Wave point set map and the video.

**Figure 5 sensors-24-01899-f005:**
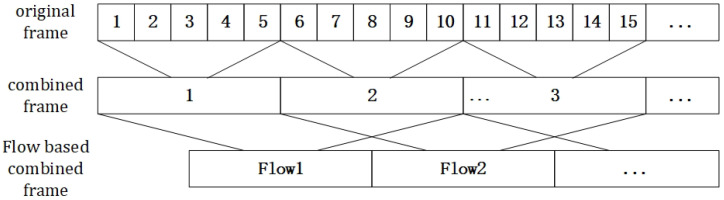
Diagram of the frame-merging process.

**Figure 6 sensors-24-01899-f006:**
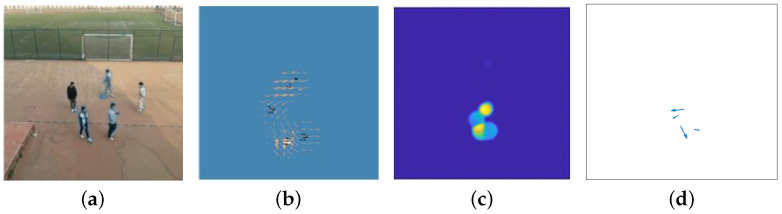
The process of local maximum sampling for RPF. (**a**) Motion scene; (**b**) RPF; (**c**) RPF scalar; (**d**) downsampled RPF.

**Figure 7 sensors-24-01899-f007:**
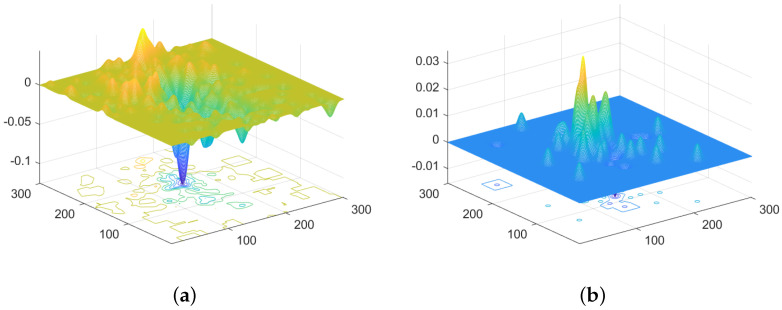
The potential energy maps for two types of crowd movements. (**a**) Crowd gathering; (**b**) crowd dispersal.

**Figure 8 sensors-24-01899-f008:**
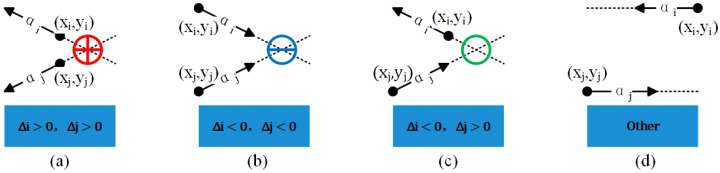
Principle of MS generation. Among them, (**a**–**d**) correspond to four different vector position relationships.

**Figure 9 sensors-24-01899-f009:**
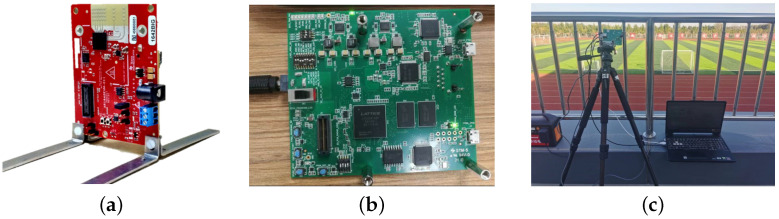
The experimental setup for data collection. (**a**) IWR1642BOOST; (**b**) DCA1000EVM; (**c**) the experimental setup with the connection configuration completed.

**Figure 10 sensors-24-01899-f010:**
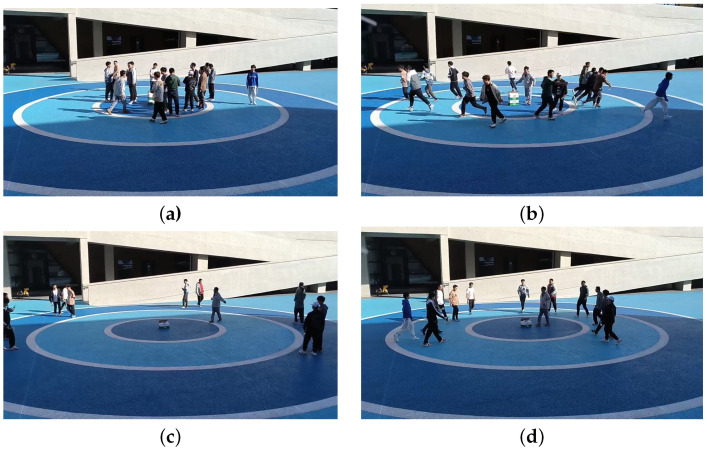
Experiment Scene 1 of (**a**,**b**) diffusion; (**c**,**d**) aggregation.

**Figure 11 sensors-24-01899-f011:**
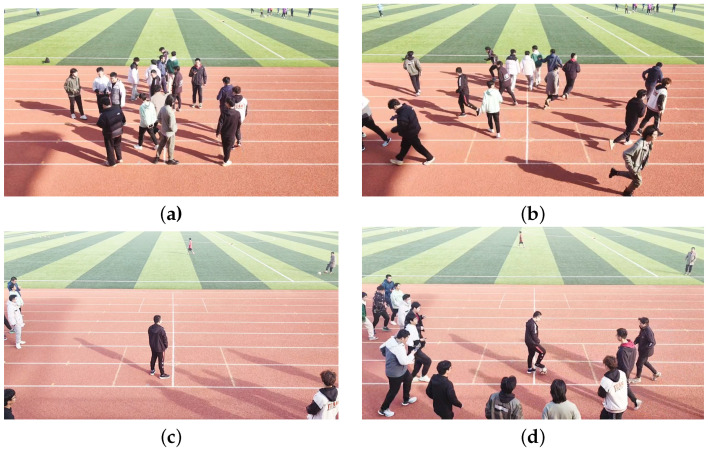
Experiment scene 2 of diffusion and aggregation, (**a**,**b**): diffusion, (**c**,**d**): aggregation.

**Figure 12 sensors-24-01899-f012:**
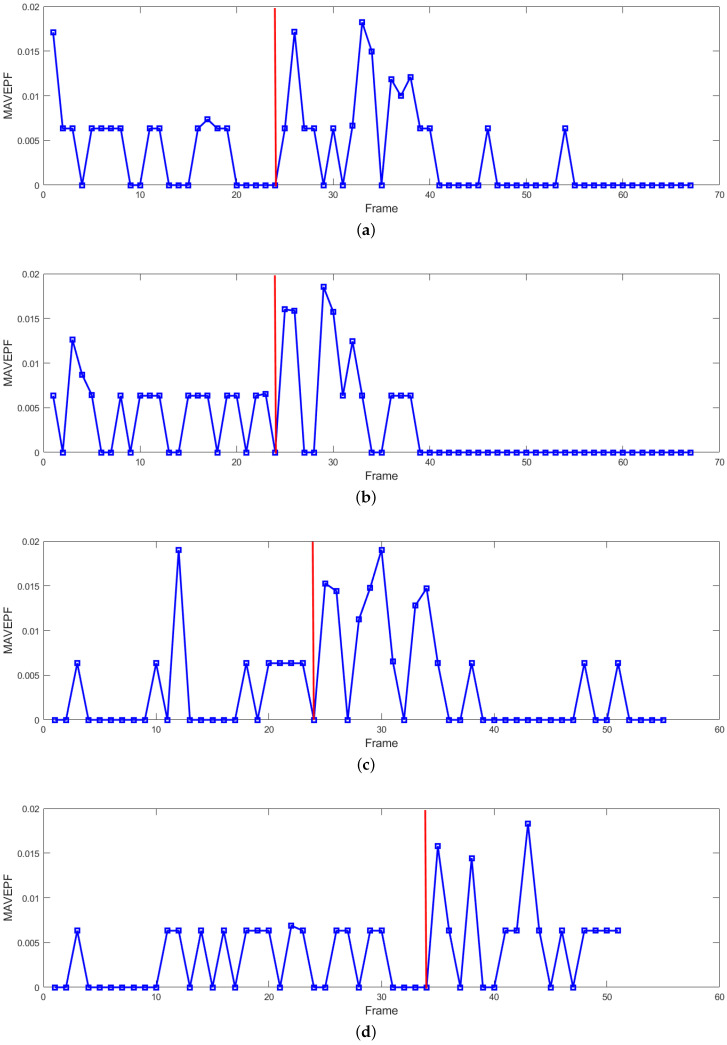
MAVEPF curve of crowd diffusion and aggregation. (**a**) Crowd diffusion in Scene 1; (**b**) crowd aggregation in Scene 1; (**c**) crowd diffusion in Scene 2; (**d**) crowd aggregation in Scene 2.

**Figure 13 sensors-24-01899-f013:**
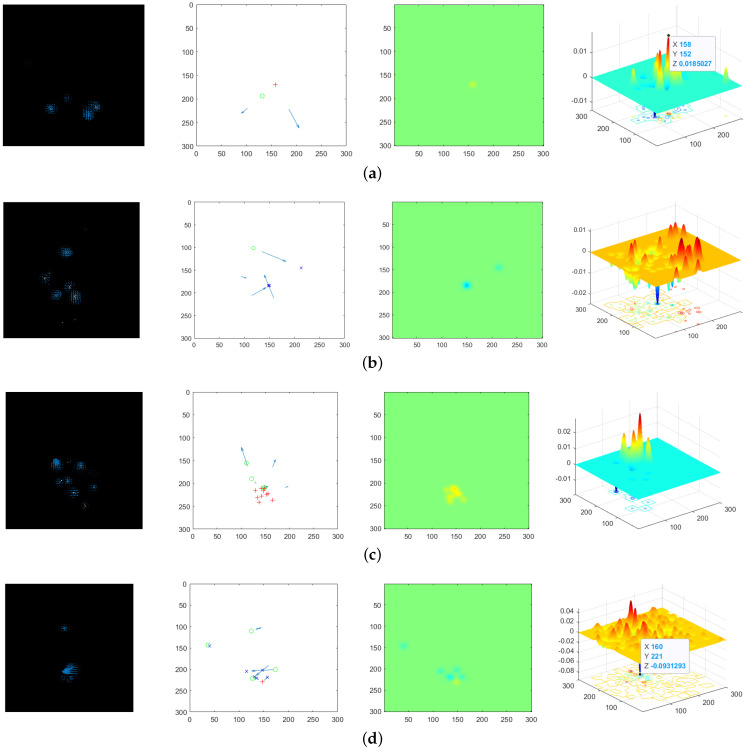
The effectiveness of the algorithm’s execution. The first column represents the RPF extracted from the point set pixel video. The second column displays the downsampled RPF, where MS is marked with red and blue. The third column shows the PFDP plane map for adjacent frames. The fourth column exhibits the accumulated PFDP surface map up to the current frame. (**a**) Crowd diffusion in Scene 1; (**b**) crowd aggregation in Scene 1; (**c**) crowd diffusion in Scene 2; (**d**) crowd aggregation in Scene 2.

**Figure 14 sensors-24-01899-f014:**
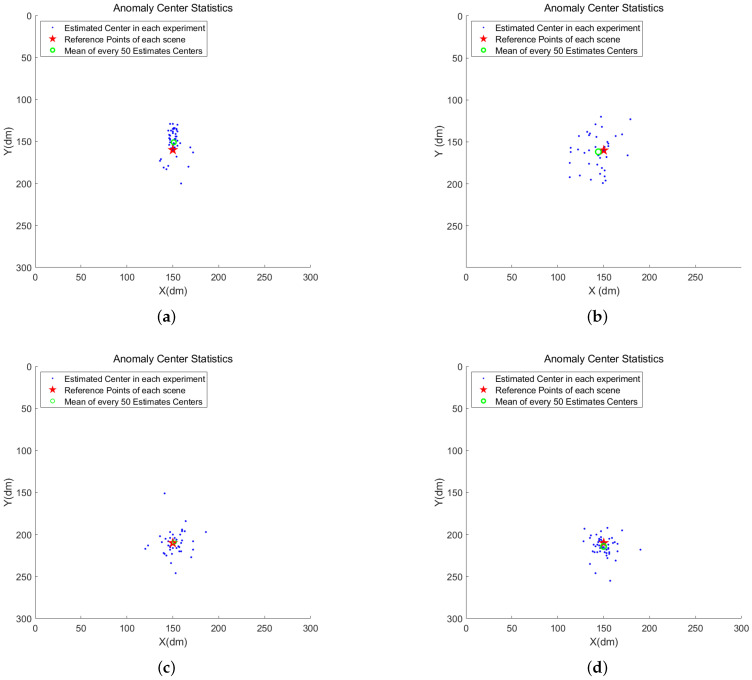
Statistics for the centers of locating sudden diffusion and aggregation using the proposed method. (**a**) Diffusion 1; (**b**) aggregation 1; (**c**) diffusion 2; (**d**) aggregation 2.

**Figure 15 sensors-24-01899-f015:**
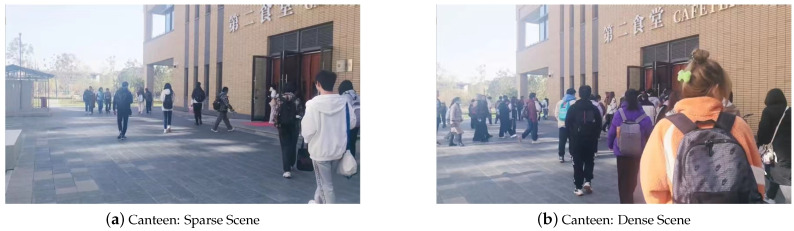
The experimental setup at the canteen entrance, where the crowd in (**b**) was denser than in (**a**).

**Figure 16 sensors-24-01899-f016:**
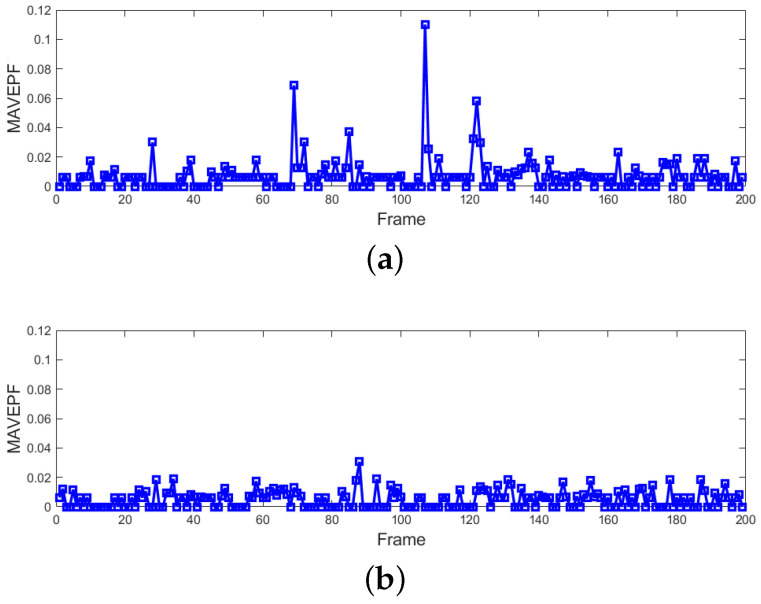
The running results of the proposed method applied to the radar data of two groups of people walking normally. (**a**) MAVEPF of Canteen Spares; (**b**) MAVEPF of Canteen Dense.

**Figure 17 sensors-24-01899-f017:**
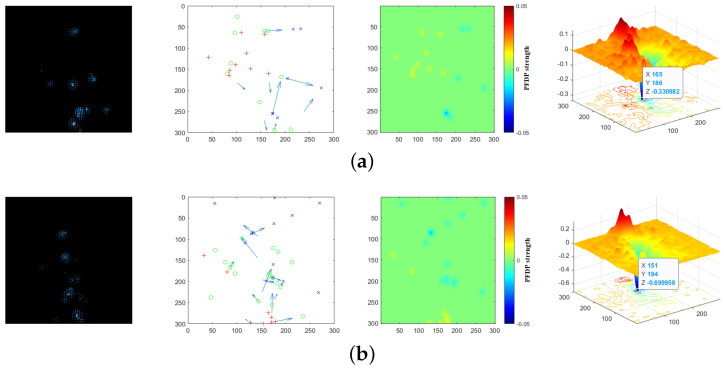
The running results of the proposed method applied to the radar data of two groups of people walking normally. Among them, (**a**) corresponds to the sparse scene in Figure 15a, and (**b**) corresponds to the dense scene in Figure 15b.

**Figure 18 sensors-24-01899-f018:**
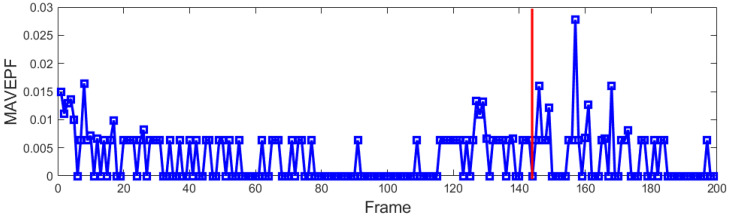
MAVEPF of roller skating training class.

**Figure 19 sensors-24-01899-f019:**
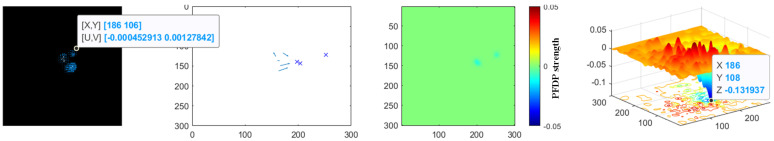
The running results of the proposed method applied to the radar-obtained data of roller skating training class.

**Figure 20 sensors-24-01899-f020:**
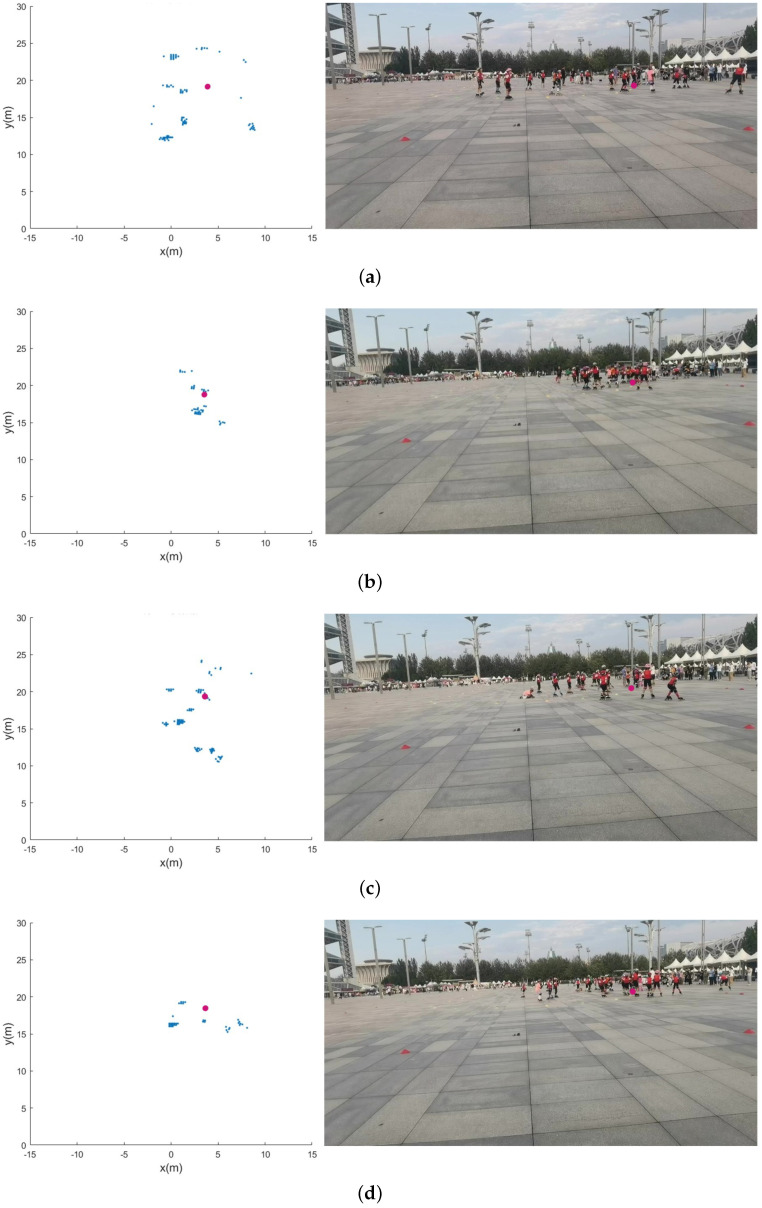
Random scene of roller skating training class. The figure shows the radar point sets and corresponding photos at four different time points. (**a**) Radar point set and video of the first aggregation; (**b**) radar point set and video of the first aggregation; (**c**) radar point set and video of the second aggregation; (**d**) radar point set and video of the second aggregation.

**Figure 21 sensors-24-01899-f021:**
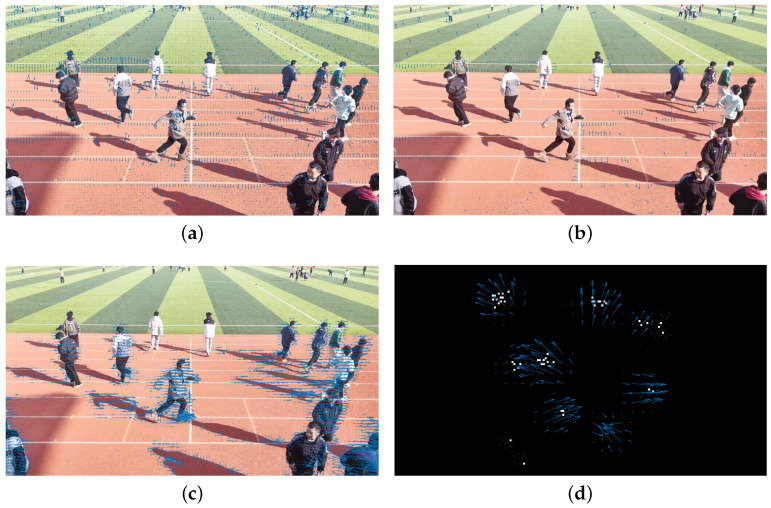
The effectiveness of three optical flow methods in crowd movement videos. (**a**) HS method; (**b**) LK method; (**c**) Farneback method; (**d**) RPF.

**Figure 22 sensors-24-01899-f022:**
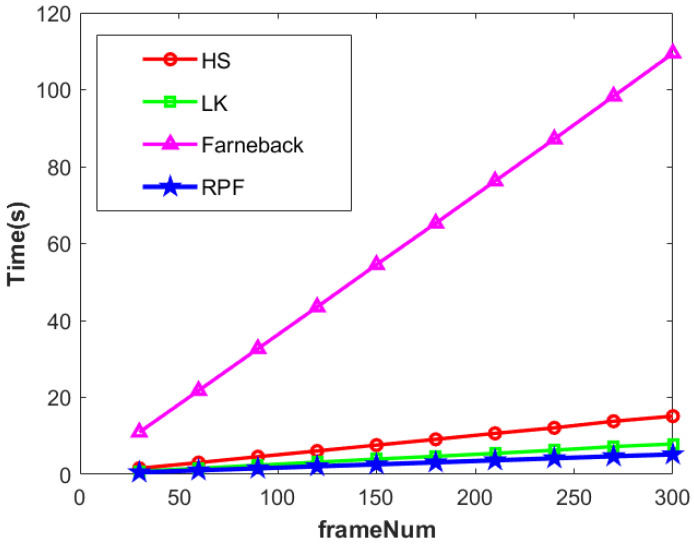
The runtime of three optical flow methods and RPF.

**Figure 23 sensors-24-01899-f023:**
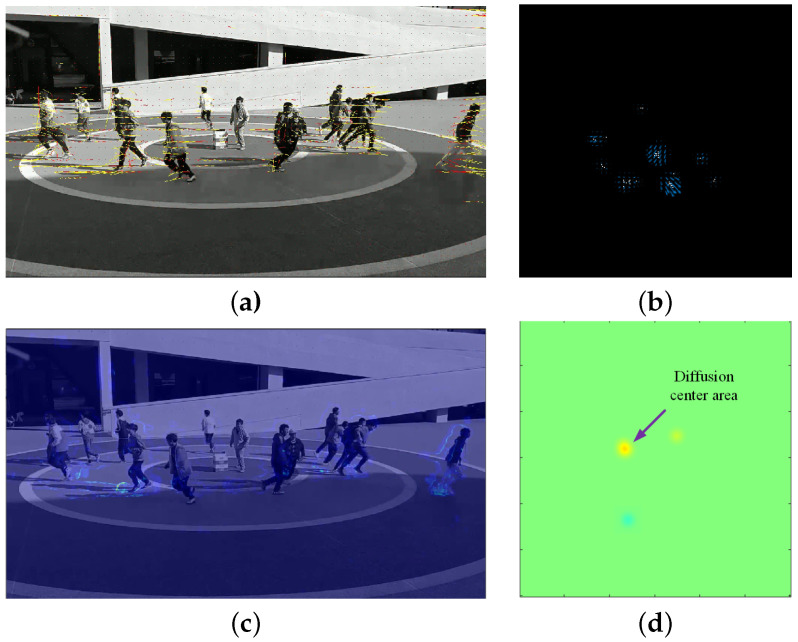
Crowd diffusion in Scene 1. (**a**) Optical flow (yellow) and SFF (red) based on video; (**b**) RPF; (**c**) ROI of SFM based on video; (**d**) ROI of PFDP.

**Figure 24 sensors-24-01899-f024:**
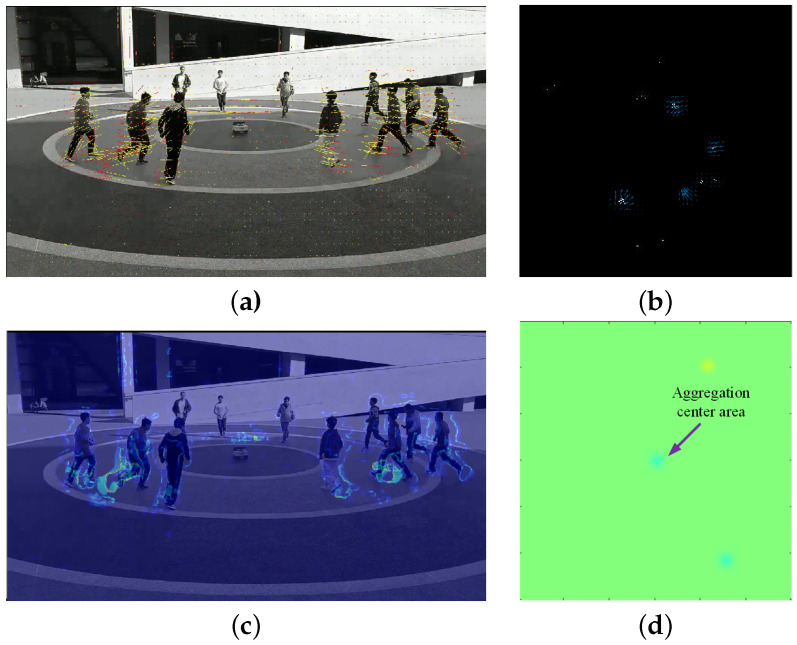
Crowd aggregation in Scene 1. (**a**) Optical flow (yellow) and SFF (red) based on video; (**b**) RPF; (**c**) ROI based on social forces; (**d**) ROI based on PFDP.

**Figure 25 sensors-24-01899-f025:**
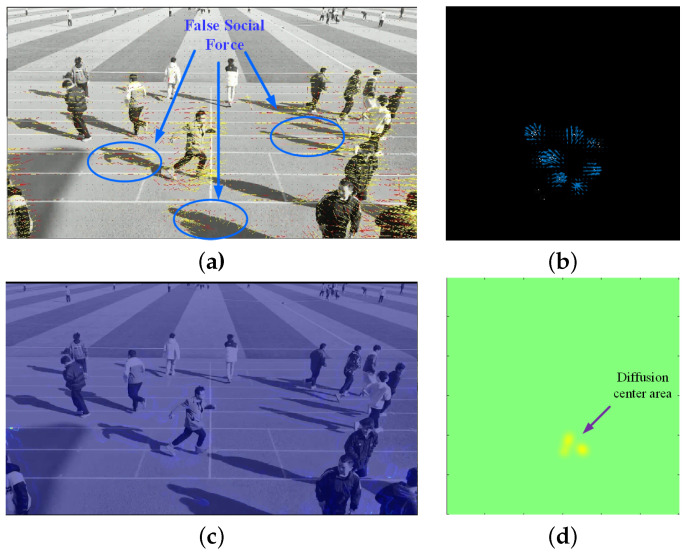
Crowd diffusion in Scene 2. (**a**) Optical flow (yellow) and SFF (red) based on video; (**b**) RPF; (**c**) ROI based on social forces; (**d**) ROI based on PFDP.

**Figure 26 sensors-24-01899-f026:**
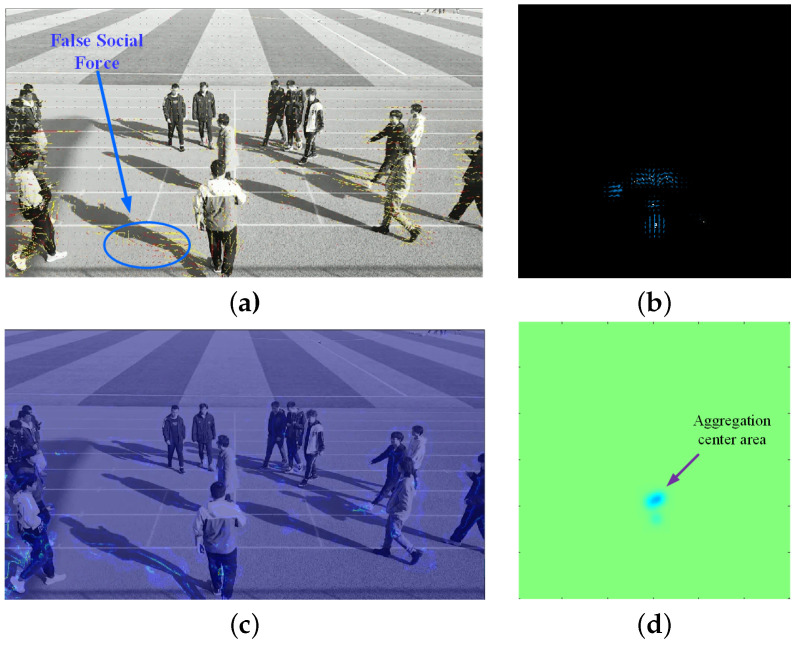
Crowd aggregation in Scene 2. (**a**) Optical flow (yellow) and SFF (red) based on video; (**b**) RPF; (**c**) ROI based on social forces; (**d**) ROI based on PFDP.

**Table 1 sensors-24-01899-t001:** Radar parameters for the experiment.

Rader Parameters	Value
Transmit antennas	2
Receive antennas	4
Starting frequency	77 GHz
Stop frequency	80.2 GHz
Bandwidth	3.2 GHz
Frequency slop	100 MHz/usec
Frame periodicity	100 msec
Chirps per frame	64
Sampling rate	6 Msps
Samples per chirp	192

**Table 2 sensors-24-01899-t002:** Experimental statistics.

Scene/Reference Points	Number of Experiments	Number of Correct PFDP Main Peak	Accuracy of PFDP Main Peak	Mean Coordinates of Localization	MAE (Meters)
Diffusion 1/(150, 160)	50	48	96%	(150.96, 150.94)	1.7
Aggregation 1/(150,160)	50	44	88%	(144.1, 161.78)	1.9
Diffusion 2/(150,210)	50	45	90%	(151.84, 208,84)	1.42
Aggregation 2/(150,210)	50	47	94%	(149.8, 214.14)	1.34

**Table 3 sensors-24-01899-t003:** Farneback parameters applicable to RPF.

Parameter	Explanation	Value
NPL	number of layers of the pyramid	4
NI	calculate number of iterations	30
NS	neighborhood size	9
FS	size of smoothing filter window	25

**Table 4 sensors-24-01899-t004:** Runtime of SFM and proposal for 100 Fframes.

Method	Diffusion in Scene 1	Aggregation in Scene 1	Diffusion in Scene 2	Aggregation in Scene 2
SFM	2875.6 s	2850.3 s	2905.8 s	2864.2 s
Ours	3.05 s	3.02 s	3.06 s	3.02 s

## Data Availability

Data are contained within the article.

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
