# Peer review of "A Crowd Movement Analysis Method Based on Radar Particle Flow"

_sensors, 2024, doi:10.3390/s24061899_

Round 1

Reviewer 1 Report

Comments and Suggestions for Authors

This paper is aimed at the research of crowd movement in the field of public safety, which has definite practical value. Because radar measurement can work all-weather and provide angle information, which is more advantageous to the analysis of crowd motion, a method of crowd motion analysis based on radar particle flow is proposed in this paper, a new concept of micro-source is defined to describe whether any two RPF vectors come from or reach the same position. The experimental results show that the proposed algorithm can effectively identify crowd gathering and dispersing motion, and the accuracy is not less than 88% , the event source is located to within 2 meters. The thesis writing logic is clear, the theory analysis is detailed, is helpful to the researcher to this kind of question thorough research. Some suggestions: 1. Supplement the sketch of coordinate system to clearly show the position relation between radar and measured object. 2. Explain how the radar acquires three-dimensional data. 3. Compare and analyze the measurement effect between the algorithm and other techniques.

Author Response

Dear reviewer, thank you very much for your valuable feedback. We have revised the manuscript according to your suggestions and have attached the questions and responses you raised. Please see the attachment.

Reviewer 2 Report

Comments and Suggestions for Authors

The authors proposed a Crowd Movement Analysis Method Based on Radar Particle Flow, which is a very interesting topic. However, the manuscript presentation is poor.
For example, many abbreviations are shown without a definition, or the definition comes late in the text. (please check the attached file)

One important comment is that the proposed system result should be compared with accuracy and computational complexity benchmarks.

Comments on the Quality of English Language

Moderate editing of English language required

Author Response

(The authors gave the same response as above.)

Reviewer 3 Report

Comments and Suggestions for Authors

In this paper, a crown movement analysis method based on radar data is proposed. The topic is interesting and applied. A comprehensive literature review is presented. But there are some grammatical errors that must be removed. Also, the structure of the manuscript must be carefully revised. For example, a section about methodology must be used and please provide subjects in a regular formwork. Please use a workflow to describe steps of this study. Also, it seems to me that results and method are mixed in some parts. Please convert name of ‘proposed’ to methodology section. Moreover, the methodology section can be presented briefly. Finally, it is strongly recommended to compare results of the proposed method with other related ones.  

Author Response

(The authors gave the same response as above.)

Reviewer 4 Report

Comments and Suggestions for Authors

Dear Authors, 

Review for the Paper title: A Crowd Movement Analysis Method Based on Radar Particle Flow

The paper looks very good in terms of organization, novelty, sufficient content, and structure. However, the authors need to consider the following points to improve the paper:

1.      Please add more literature review related to the topic and Radar

2.      Add label to the color bars in Figures 17 and 19

3.      Add more explanation to Figures 18, 19 and 20.

4.      Please consider to discuss your results in reference to other people work in literature.

All the best

Author Response

(The authors gave the same response as above.)

Round 2

Reviewer 2 Report

Comments and Suggestions for Authors

The authors did a great effort in the manuscript revised version and they fulfilled all of my comments . Thanks

Reviewer 3 Report

Comments and Suggestions for Authors

-